# Intraocular Pressure Measurement in Childhood Glaucoma under Standardized General Anaesthesia: The Prospective EyeBIS Study

**DOI:** 10.3390/jcm11102846

**Published:** 2022-05-18

**Authors:** Alicja Strzalkowska, Nina Pirlich, Julia V. Stingl, Alexander K. Schuster, Jasmin Rezapour, Felix M. Wagner, Justus Buse, Esther M. Hoffmann

**Affiliations:** 1Department of Ophthalmology, University Medical Centre of the Johannes Gutenberg, University Mainz, 55131 Mainz, Germany; alicja.strzalkowska@unimedizin-mainz.de (A.S.); julia.stingl@unimedizin-mainz.de (J.V.S.); alexander.schuster@uni-mainz.de (A.K.S.); jasmin.rezapour@unimedizin-mainz.de (J.R.); felix.wagner@unimedizin-mainz.de (F.M.W.); jbuse@students.uni-mainz.de (J.B.); 2Department of Anaesthesiology, University Medical Centre of the Johannes Gutenberg, University Mainz, 55131 Mainz, Germany; pirlich@uni-mainz.de

**Keywords:** childhood glaucoma, intraocular pressure measurement, iCare tonometry, Perkins tonometry, standardized anaesthesia

## Abstract

Objective: We aimed to compare intraocular pressure (IOP) measurements using iCare^®^ PRO rebound tonometry (iCare) and Perkins applanation tonometry (Perkins) in childhood glaucoma subjects and healthy children and the influence of anaesthesia depth, age and corneal thickness. Material: Prospective clinical, case-control study of children who underwent an ophthalmologic examination under general anaesthesia according to our protocol. Children were 45.45 ± 29.76 months old (mean ± SD (standard deviation)). Of all children, 54.05% were female. IOP was taken three times (T1–T3), according to duration and the depth of anaesthesia. The order of measurement alternated, starting with iCare. Agreement between the device measurements was evaluated using Bland–Altman analysis. Results: 53 glaucoma subjects and 22 healthy controls. Glaucoma subjects: IOP measured with iCare was at T1: 27.2 (18.1–33.8), T2: 21.6 (14.8–30.6), T3: 20.4 mmHg (14.5–27.0) and Perkins 17.5 (12.0–23.0), 15.5 (10.5–20.5), 15.0 mmHg (10.5–21.0) (median ± IQR (interquartile range)). Healthy controls: IOP with iCare: T1: 13.3 (11.1–17.0), T2: 10.6 (8.1–12.4), T3: 9.6 mmHg (7.7–11.7) and Perkins 10.3 (8.0–12.0), 7.0 (5.5–10.5), 7.0 mmHg (5.5–8.5) (median ± IQR). The median IOP was statistically significantly higher with iCare than with Perkins (*p* < 0.001) in both groups. The mean difference (iCare and Perkins) was 6.0 ± 6.1 mmHg for T1–T3, 7.3 at T1, 6.0 at T2, 4.9 mmHg at T3. Conclusion: The IOP was the highest in glaucoma subjects and healthy children at T1 (under sedation), independently of the measurement method. iCare always leads to higher IOP compared to Perkins in glaucoma and healthy subjects, regardless of the duration of anesthesia.

## 1. Introduction

Childhood glaucoma is a rare disease, with incidence in Europe of 1 per 20,000 live births [1]. If undiagnosed and consecutively treated too late, this disease can result in visual impairment or blindness in 1.2 to 7.1%, depending on the country of origin [2,3,4]. Early and accurate diagnosis in childhood glaucoma is crucial to initiate an appropriate therapy. This prevents irreversible damage to the cornea, optic nerve, as well as development of buphthalmos and myopia with vision loss [5,6,7].

To diagnose glaucoma in children, the ophthalmological examination with evaluation of ocular dimensions, corneal clarity, optic nerve and intraocular pressure measurement is needed [8]. However, the clinical examination can be challenging in uncooperative children [8]. Distorting factors, such as children crying, eyes squeezing or intrathoracic pressure may lead to inaccurate measurements. That is the reason why the success rate of the intraocular pressure (IOP) measurement in awake children varies between 14–60% [9,10]. To exclude these influencing factors, the necessary ophthalmological examination needs to be performed under general anaesthesia [8]. The general anaesthesia may affect the IOP itself, depending on the given sedatives, depth of anaesthesia or usage of anaesthetic techniques, such as laryngoscopy or intubation [11]. For instance, ketamine and suxamethonium increase the IOP [11,12,13], while remifentanil decreases IOP [14,15,16]. The increasing depth of anaesthesia leads to a significant reduction in the IOP [17].

Moreover, depending on the selected IOP measurement method itself, the IOP values can vary. The iCare^®^ PRO rebound tonometry (iCare) is easy to use and does not need eye drops to carry out the measurement. However, IOP values measured with iCare, depending on device generation, are affected by corneal thickness and differ in sitting or supine position [18,19,20]. Age can also affect the measurement with iCare due to age-related changes in collagen fibrils in the cornea, which lead to an increase in stiffness [21,22]. To achieve reliable values with Perkins applanation tonometry (Perkins), on the other hand, some practice is required [23]. To perform the measurement with this device, fluorescein/anesthetic eye drops are needed. According to Garcia et al., the iCare measurement overestimates the IOP compared to Perkins tonometry [17]. However, Molero-Senosiaín proved that the iCare overestimates only the high IOP in comparison to Perkins [17].

Our main goal must be a precise and reliable IOP measurement in children under anaesthesia, as close as possible to the awake state without relevant changes in IOP [10,11]. To achieve that, the relationship between depth of anaesthesia and IOP has to be investigated [15,24].

In our study, the IOP and central corneal thickness (CCT) of childhood glaucoma subjects and healthy children was performed under protocol-defined standardized general anaesthesia. The protocol was established in our Childhood Glaucoma Centre at the University Medical Center in Mainz, Germany [25].

The purpose of this study was to compare IOP measurements using iCare and Perkins in childhood glaucoma subjects and healthy children at different time points of anaesthesia.

In addition, the correlation between CCT and IOP measurements, as well as between age and IOP, obtained with both devices, was analyzed.

## 2. Materials and Methods

### 2.1. Study Design

This study was approved by the local Ethics Committee of the Medical Association of the Rhineland-Palatinate state, Germany (Approval number: 2019-14207). This was a single centre, prospective cohort study of all childhood glaucoma subjects (53) who underwent an ophthalmologic examination including IOP and CCT measurement under protocol-defined standardized general anaesthesia between April 2019 and March 2021 at the University Eye Hospital Mainz, Germany. IOP was taken at three predefined time points (T1–T3) according to the depth of anaesthesia. The time of measurement was the same for each and every child. A precise description of the measurement can be found below under ‘sequence of measurement’. Twenty-two children without a history of glaucoma were included as a control group. The correlation between iCare and Perkins was the primary endpoint. The IOP, the correlation between IOP and CCT, and IOP and age, were secondary endpoints.

### 2.2. Intraocular Pressure Measurement

IOP was measured with iCare^®^ PRO rebound tonometry (iCare, Tiolat Oy, Helsinki, Finland) and Perkins applanation tonometry (Clement Clarke, Haag-Streit, Harlow, United Kingdom). The development of the rebound tonometry, originating from Kontiola, in 2001 led to iCare measurement. The magnetized probe launches against the eye using a solenoid. The solenoid captures the movement and impact of the probe on the eye [26].

The Perkins tonometer shares the same principle used in Goldmann tonometry. It is based on Imbert–Fick law. The force required to cover the area of a sphere to applanate is exactly the same size as the pressure inside the sphere and the applanated area [27]. Both tonometers are portable devices. The measurements were carried out by one of four ophthalmologists specialized in glaucoma in our clinic with wide experience in this field. A series of measurements was carried out on each child by the same specialist. All measurements were taken in a horizontal position.

### 2.3. Sequence of Measurements

IOP was measured in both eyes at three times (T1–T3). The first IOP measurement was performed immediately after the application of the propofol bolus (stage 1, T1). At this point the child was spontaneously breathing, slightly sedated, titrated with a maximum of 4 mg/kg bodyweight propofol intravenous bolus. This measurement reflects most closely the state of consciousness. The second IOP measurement was performed one minute after insertion of the laryngeal mask (stage 2, T2). At that point, a larger (anesthetic dose) bolus of propofol was given. The propofol and remifentanil were also running as perfusors. It means at this point the child was in a very deep anaesthesia. After that, the laryngeal mask was blocked according to the manufacturer’s instructions and with the aid of the cuff pressure gauge to max. 60 cm H_2_O. Immediately after the blocking, the third IOP measurement was acquired (stage 3, T3). At this time, the depth of anesthesia is approximately the same as at T2. At each stage, iCare measurement was followed by the measurement with Perkins, see Figure 1.

### 2.4. Corneal Thickness Measurement

The corneal thickness was measured with ultrasound pachymetry (Tomey AL-3000 (Tomey, Nuremberg, Germany).

### 2.5. Inclusion Criteria

Children who met the following criteria were eligible for this study: indication for general anaesthesia with laryngeal mask for an operative or diagnostic intervention, age from 0.5 to 10 years, 1–3 according to the American Society of Anaesthesiologists physical status classification system (ASA classification), present written declaration of consent of the legal representatives.

### 2.6. Exclusion Criteria

Contraindications for the use of a laryngeal mask, known allergy to propofol or remifentanil was an exclusion criterion.

### 2.7. Childhood Glaucoma Subjects

To define childhood glaucoma, we used the Childhood Glaucoma Research Network criteria such as: IOP > 21 mmHg, optic disc cupping, corneal findings (Haab striae, Diameter > 11 mm in newborn, >12 mm in child < 1 year of age > 13 mm any age), progressive myopia/myopic shift, reproducible visual field defect which could not be caused by another reason. To meet the definition, at least two criteria have to be fulfilled [6].

### 2.8. Healthy Subjects

Those children needed an operation due to strabismus or tear duct obstruction. The children were otherwise healthy and did not require continuous local or systemic medication.

### 2.9. Statistical Analysis

Categorical variables were presented as frequencies with percentages, whereas median and interquartile range (IQR) or mean ± standard deviation (SD) were used to describe continuous variables. Evaluation of data normality was performed using Shapiro–Wilk test, whereas variance equality was verified by Levene’s test. Normally distributed variables were compared using the *t* test. Non-normally distributed continuous variables were analysed using Friedman test, Wilcoxon test and Spearman’s rank correlation coefficient. For multiple comparisons, Bonferroni correction was applied. Categorical variables were compared using the χ^2^ test. Analyses were performed using R version 4.0.5 and Statistica 13.1 software for Windows. All statistical tests were two-tailed with the significance level set at α = 0.05. Data were compared by determining interclass correlation coefficients for each tonometer and representing the differences detected as Bland–Altman plots.

## 3. Results

### 3.1. Characteristics

In this study, 150 eyes of 75 children were included; that is, 53 glaucoma subjects and 22 healthy controls. Overall age was 45.5 ± 29.8 months, glaucoma subjects 46.2 ± 29.7 and healthy children 43.8 ± 30.5 months old (mean ± SD). Of all the children, 54% were female. The mean CCT for the glaucoma subjects was statistically significantly higher than for the healthy ones, 601.6 ± 104.7 µm vs. 554.5 ± 39.7 µm, respectively (*p* = 0.009). The range of CCTs measured was 334.0–818.5 µm for glaucoma subjects and 494.0–616.5 µm for healthy children, see Figure 2. Patient 29 was excluded from the analysis because the CCT showed an extremal outlier from others, as a result of a massive corneal oedema.

### 3.2. Primary Endpoint—Correlation between iCare and Perkins

The correlation between iCare and Perkins for all children was moderate: the intraclass correlation coefficient (r) between the two methods was at T1–T3 0.63 (*p* < 0.001), at T1: 0.54 (*p* < 0.001), at T2: 0.62 (*p* < 0.001), and at T3: 0.72 (*p* < 0.001). The mean difference (iCare and Perkins) was 6.0 ± 6.1 mmHg for T1–T3, 7.3 mmHg at T1, 6.0 mmHg at T2, 4.9 mmHg at T3, see Figure 3. 

### 3.3. Secondary Endpoints

#### 3.3.1. Median IOP with iCare and Perkins

For all children, the median IOP measured with iCare was statistically higher than measured with Perkins (*p* < 0.001). The median IOP using iCare was at T1: 20.3 (13.1–30.3), T2: 15.5 (11.0–25.2), T3: 15.7 mmHg (10.1–25.1) and for Perkins at T1: 14.5 (9.5–20.5), T2: 11.0 (7.5–19.5), T3: 11.5 mmHg (7.5–19.0) (median ± IQR). In addition, it was shown that IOP values measured with iCare and Perkins significantly decreased over time—median IOP at T1 > T2 (*p* < 0.001) and T1 > T3 (*p* < 0.001) for both devices.

For glaucoma subjects, the median IOP measured with iCare was statistically higher than measured with Perkins (*p* < 0.001). The median IOP measured with iCare was at T1: 27.2 (18.1–33.8), T2: 21.6 (14.8–30.6), T3: 20.4 mmHg (14.5–27.0) and Perkins T1: 17.5 (12.0–23.0), T2:15.5 (10.5–20.5), and T3: 15.0 mmHg (10.5–21.0) (median ± IQR). In addition, it was shown that IOP values in the iCare measurement significantly decreased over time between T1 and T2 and T1 and T3 and did not change between T2 and T3—median IOP at T1 > T2 (*p* < 0.001), T2 = T3 (*p* = 0.101), and T1 > T3 (*p* < 0.001). With Perkins, IOP values fell between T1 and T2 (*p* < 0.003). The IOP did not change significantly between T2 and T3 (*p* = 0.976) and T1 and T3 (*p* < 0.022), see Figure 4a. The IOP reduction between T1 and T2 is 21% for iCare and 11% for Perkins.

For healthy subjects, the median IOP measured with iCare was statistically higher than measured with Perkins (*p* < 0.001). The median IOP for iCare at T1 was: 13.3 (11.1–17.0), T2: 10.6 (8.1–12.4), T3: 9.6 mmHg (7.7–11.7) and for Perkins at T1: 10.3 (8.0–12.0), T2: 7.0 (5.5–10.5), and T3: 7.0 mmHg (5.5–8.5) (median ± IQR). In addition, it was shown that IOP values in the iCare measurement decreased significantly over time between T1 and T2 (*p* < 0.001) and T1 and T3 (*p* < 0.001) was not changed between T2 and T3—median IOP at T1 > T2 (*p* < 0.001), T2 = T3 (*p* = 0.101), and T1 > T3 (*p* < 0.001). With Perkins, IOP values fell between T1 and T2 (*p* < 0.001) and T1 and T3 (*p* < 0.001). The IOP did not change significantly between points T2 and T3—median IOP at T1 >T2 (*p* < 0.001), T2 = T3 (*p* = 0.015), and T1 > T3 (*p* < 0.001), see Figure 4b. The IOP reduction between T1 and T2 was 20% for iCare and 32% for Perkins.

#### 3.3.2. Correlation of CCT and IOP

As for the correlation between CCT and IOP: the trend is downward in healthy children, but not statistically significant in iCare (*p* = 0.837) and not statistically significant in Perkins (*p* = 0.656). In glaucoma subjects, the trend is increasing, but is not statistically significant in iCare (*p* = 0.228) and borderline in Perkins (*p* = 0.057), see Figure 5.

#### 3.3.3. Correlation of Age and IOP

The correlation between age and IOP with iCare at T1 was weak positive (*p* = 0.009) in glaucoma subjects and not statistically significant in healthy subject (*p* = 0.243). The correlation between age and IOP with Perkins was very weak positive (*p* = 0.082) in glaucoma subjects and not statistically significant in healthy subjects (*p* = 0.263), see Figure 6.

## 4. Discussion

In this study, we compared IOP measured with iCare and Perkins in childhood glaucoma subjects and healthy children and the influence of anaesthesia depth, age and corneal thickness.

In childhood glaucoma, a prompt and accurate diagnosis should be our main objective to avoid irreversible eye damage that can lead to blindness.

Glaucoma detection can be difficult for various reasons. It is rare that children experience the entire range of glaucoma characteristics, such as buphthalmos, corneal clouding or symptoms like epiphora, without obstruction of the tear ducts. It is common that an ophthalmological examination is not possible. In children, the vision field examination is difficult to carry out because of a lack of cooperation or refusal of eye patch [28,29,30,31].

This illustrates the importance of a precise and reliable IOP measurement in diagnosing childhood glaucoma [11]. To achieve it, it is necessary to ensure optimal test conditions. Most often, in smaller children, it is only possible under general anaesthesia.

However, until now, there has been a lack of standardized anaesthesia protocol in healthy children and children with glaucoma, which would accurately determine the type and amount of drug administered, anaesthetic procedures, such as mask application, mask blocking or intubation, depth of anaesthesia, as well as the best moment to perform the IOP measurement, as well as the optimal type of IOP measuring device must be determinate. Our study protocol was published recently [25]. In this paper, we want to present the data obtained using the described ophthalmological–anaesthetic protocol *EyeBIS*.

Our single centre, prospective, standardised cohort study, included, in total, 75 glaucoma subjects and healthy children. To the best of our knowledge, this is the first study where the IOP was measured in children during standardized general anaesthesia. Only very few studies have been published regarding IOP measurement with different devices in childhood glaucoma [32,33,34,35,36].

According to our study, the correlation between iCare and Perkins for all children was moderate for all three time points altogether. In the course of anaesthesia, lower IOP values were measured. The lower the IOP values, the better the correlation between the two measurement methods, T1: 0.54, T2: 0.62, and T3: 0.72. There are not many papers concerning correlation coefficients between those two devices in children, but there is even less information in this regard in glaucoma children. Martinez-de-la-Casa et al. reported a correlation coefficient of 0.87 in childhood glaucoma [20]. However, the mean IOP in this study was 22.1 ± 7.7 mm Hg for iCare and 19.1 ± 5.4 mmHg for Perkins. In our study, the IOP measured with iCare was 26.8 ± 11.2 at T1; 23.2 ± 11.4 at T2; 21.7 ± 10.3 at T3 and with Perkins 18.2 ± 7.7 at T1; 16.0 ± 7.1 at T2; 16.1 ± 7.4 at T3. Children in the study of Martinez-de-la-Casa et al. were older. They were 8.8 ± 2.9 years old, whereas in our study, the children were 45.5 ± 29.8 months old [20]. It is conceivable that the corneal characteristics and, hence, the IOP measurement in these two groups were different.

The median IOP measured with iCare was statistically higher compared with Perkins for glaucoma subjects and healthy children in our study. Our results confirm some earlier findings from the study by Borrego et al. [32]. In this study, the IOP was higher with iCare than with Perkins in the glaucoma children as well. In contrast to our study, Borrego et al. did not find a statistically significant mean IOP difference between iCare and Perkins, (0.42 ± 3.69 mmHg, *p* = 0.41) [32]. In our study, the mean difference between those two devices was statistically significant: 6.0 ± 6.1 mmHg for T1–T3, 7.3 mmHg at T1, 6.0 mmHg at T2, and 4.9 mmHg at T3. There are two explanations for it. In our study, the IOP was measured in sedation. At this time, the IOP is the highest. In higher IOP, iCare tends to overestimate IOP in comparison to Goldmann tonometry [37]. Additionally, there were many high IOP values measured within this study, maximal IOP with iCare was 33.8 vs. 29.1 mmHg compared to the Borrego et al. study. Because the mean difference between iCare and Perkins varies depending on the depth of general anaesthesia, the iCare and Perkins measurements are not interchangeable and cannot be converted directly into one another.

On the contrary, there are studies that proved that both iCare and Goldmann tonometry underestimate the real IOP. According to the study by Takagi at al., iCare tends to underestimate the IOP in comparison to Goldmann tonometry in lower IOP [38]. The reason for it was not given in this paper. The latest Messenio et al. study, concerning Goldmann tonometry, described the underestimation of IOP by Goldmann tonometry due to thinner corneas, which was already mentioned in another prior study [39,40]. We cannot support these statements with our data. Our IOP measurements were high from the beginning and the corneal thickness was normal to high.

The IOP was the highest at the measurement performed immediately after the application of propofol bolus (T1), regardless of the measurement method, for all children, glaucoma subjects and healthy children. This measurement reflects most closely the state of consciousness. After that, the deep anaesthesia was provoked and the second measurement was conducted. At this time the IOP was significantly lower in comparison to T1, once again, regardless of the measurement method for all children, glaucoma subjects and healthy children. At T3, the IOP was no different in comparison to T2 in glaucoma subjects or healthy controls. This is not surprising since the depth of general anaesthesia did not change between T2 and T3. It confirms the results of the previous study of Barclay et al. [41].

Our study confirms that the general anaesthesia reduced IOP significantly [16]. That highlights, once again, how important the cooperation and communication between anaesthesiologist and ophthalmologist are. The ophthalmologist needs to be in the operating room before the beginning of any anaesthesiologic procedures. By this and a standardized anaesthetic protocol, the measured IOP is as close to real IOP/awake IOP as possible.

In contrast to the study of Muir et al., we found differences in CCT in glaucoma subjects and healthy children [42]. It could be caused by the corneal oedema by strongly elevated IOP in our study. However, we did not find a statistically significant correlation between CCT and IOP. There are other known biomechanical properties in the cornea, which can influence the IOP, such as the corneal hysteresis [43,44], the corneal visoelastic parameter [45], which should be analysed in glaucoma children in the future.

The correlation between age and IOP with iCare and Perkins at T1 was weak positive in glaucoma subjects in our study. Therefore, we could not confirm the results of Sihota et al., where an increasing IOP with age was found [9]. It is probably caused by the small number of children who represent each age range in our study.

### Strengths and Weaknesses

EyeBis has many strengths, including its prospective nature, large group of children glaucoma subjects, and standardized anaesthetic protocol for the comparison of two different IOP measurement devices. There are some limitations as well. The mean IOP was measured for both eyes of the child at each timepoint. The sequence of measuring, first iCare, then Perkins, might have, at least partly influenced the lowering of the IOP measured with Perkins. As mentioned before, the iCare probe launches against the eye using a solenoid. It could lead to aqueous massage or corneal impression. It is proven that repeated iCare readings tend to lower the IOP [36].

## 5. Conclusions

Under standardized general anaesthesia conditions, tonometry devices present differences in IOP. iCare leads to higher IOP compared to Perkins in glaucoma and healthy subjects, regardless of the duration of anesthesia.

The IOP changes during the course of anaesthesia and should be measured at the beginning of anaesthesia, according to our protocol, because at this point, the IOP is the highest. The knowledge of the exact anaesthesia depth during IOP measurement gives (a) more confidence in IOP values and (b) enables the glaucoma surgeon to interpret IOP results more accurately. In our study, IOP was independent of CCT.

## Figures and Tables

**Figure 1 jcm-11-02846-f001:**
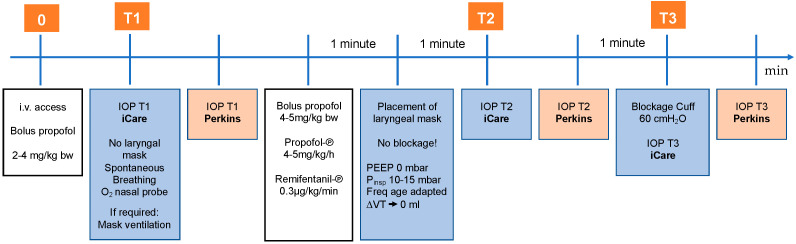
Sequence of measurements and procedures within EyeBIS study. Adapted from Pirlich, N.; Grehn, F.; Mohnke, K.; Maucher, K.; Schuster, A.; Wittenmeier, E.; Schmidtmann, I.; Hoffmann, E.M. Anaesthetic Protocol for Paediatric Glaucoma Examinations: The Prospective EyeBIS Study Protocol. BMJ Open 2021, 11, e045906 [25]. Abbreviation: i.v.: intravenous; bw: bodyweight; IOP: intraocular pressure; PEEP: Positive End-Expiratory Pressure; Pinsp: inspiratory pressure; Freq: frequency; VT: tidal volume.

**Figure 2 jcm-11-02846-f002:**
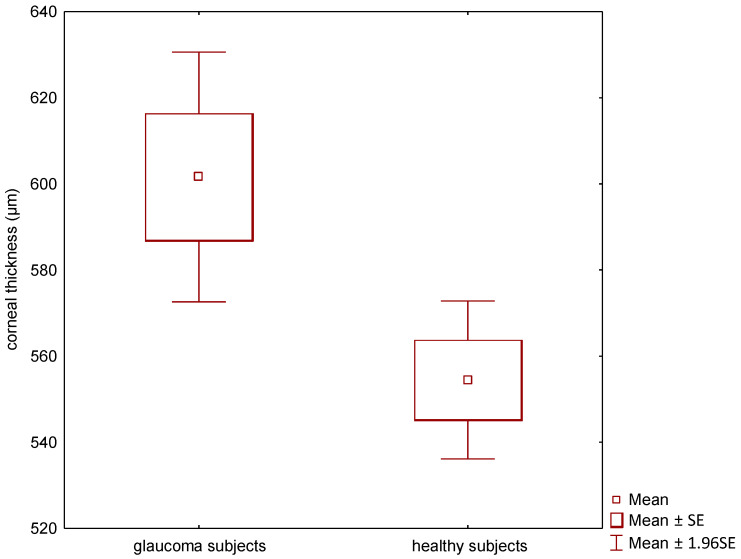
Corneal thickness for glaucoma vs. healthy subjects in the form of box plots.

**Figure 3 jcm-11-02846-f003:**
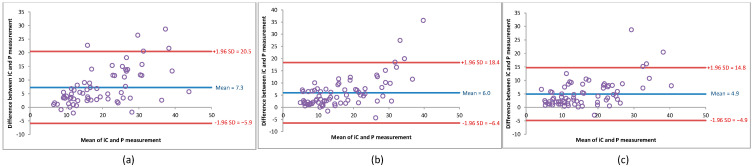
Difference between the measurement with iCare and Perkins in the form of Bland–Altman plot at the different time points for all children (**a**) T1, (**b**) T2, (**c**) T3. Abbreviation: IC: iCare; P: Perkins.

**Figure 4 jcm-11-02846-f004:**
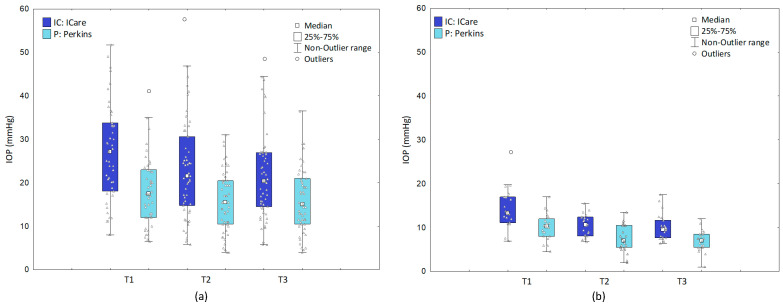
Median IOP at T1–T3 measured with iCare and Perkins (**a**) glaucoma subjects (**b**) healthy children. Abbreviation: IOP: intraocular pressure; IC: iCare; P: Perkins.

**Figure 5 jcm-11-02846-f005:**
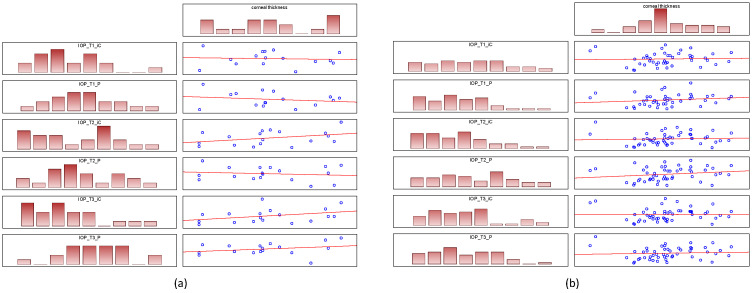
Correlation of CCT and IOP for iCare and Perkins (**a**) glaucoma subjects (**b**) healthy children. Abbreviation: CCT: corneal thickness; IOP: intraocular pressure; IC: iCare; P: Perkins.

**Figure 6 jcm-11-02846-f006:**
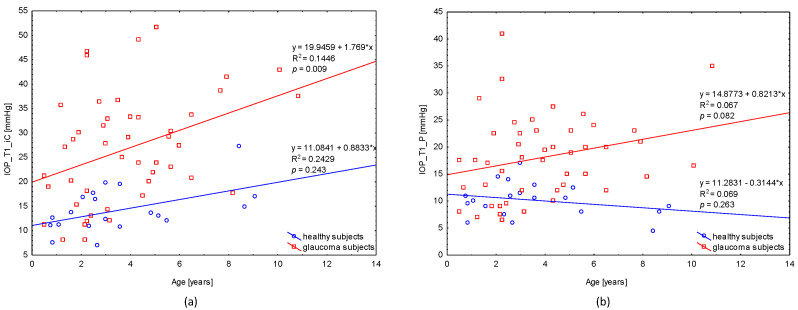
Correlation of age and IOP for (**a**) iCare and (**b**) Perkins at T1. Abbreviation: IOP: intraocular pressure; IC: iCare; P: Perkins; y: function; x: age; R^2^: regression squared error metric; *p*: *p*-Value.

## Data Availability

Not applicable.

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
