# Peer review of "Intraocular Pressure Measurement in Childhood Glaucoma under Standardized General Anaesthesia: The Prospective EyeBIS Study"

_jcm, 2022, doi:10.3390/jcm11102846_

Round 1

Reviewer 1 Report

This was an interesting paper. I feel there was room to expand and add a little more detail. Some re-wording required to improve the flow.

Author Response

Point 1: This was an interesting paper.

Response 1: Thank you very much for the detailed and helpful comments on our work.

Point 2: I feel there was room to expand and add a little more detail. Some re-wording required to improve the flow.

Response 2: We added more details and re-wrote some of the words, as you wished.

Point 3: Line 16: breakdown of participant profile eg mean age, male: female ratio.

Response 3: We added it to the manuskript: Children were 45.45 ± 29.76 months old [mean ± SD]. 54.05% of all children were female.

Point 4: Line 50: instead of rise, increase.

Response 4: The word was altered in the manuskript.

Point 5: The sentence in the line 56 and 57 could this be expanded.

Response 5: It has been implemented and the current sentence is: Age can also affect the measurement with iCare due to age-related changes in collagen fibrils of cornea which lead to increase in stiffness[21].

Point 6: Line 83, 84: what does this mean? could it be explained for the reader, were they taken at the same time points during anaesthesia for every participant?

Response 6: The following information was added to the manuskript: ‘IOP was taken at three predefined time points (T1-T3) according to the depth of anaesthesia. The time of measurement was the same for each and every child. A precise description of the measurement can be found below under 'sequence of measurement'.’

Point 7: Line 98: were they all ophthalmologists? did they all have a similar level of experience?

Response 7: The sentence was changed to include all needed iformations:  The measurements were carried out by one of 4 ophthalmologists specialized in glaucoma in our clinic with wide experience in this field.’

Point 8: Line 128: health controls needing an operation were included and so this indicates they had some sort of general health issue. Were the healthy controls on any medication that could have influenced IOP. Was this checked and listed as an exlcusion criteria?

Response 8: To answer this quastion an extra paragraph was included to the manuskript: ‘2.8. Healthy subjects Those children needed an operation due to strabismus or tear duct obstruction. The children were otherwise healthy and did not require continuous local or systemic medication.’

Point 9:  Line 188: what was the average time between T1, T2 and T3?

Response 9: We do not have exactly data on it. The first IOP measurement with iCare and Perkins was done just after the propofol bolus. After that the the laryngeal mask was imeditally inserted. One minute after this acction the second IOP measurment was done. Following this the laryngeal mask was blocked. Immediately after the blocking, the third IOP measurement was acquired.

Point 10: Line 218: image cut off from bottom

Response 10: We double checked it, but in our version it appears correctly.     

Point 11: Line 325: re-word to downward.

Response 11: We changed it to ‘lower the IOP’.

Reviewer 2 Report

The authors need to be commended for the manuscript.  It is well known that I-care tonometer tends to overestimate IOP as compared to GAT, especially at higher IOP values. The authors have compared intraocular pressure (IOP) measurements using rebound tonometry (iCare) and Perkins applanation tonometry  in childhood glaucoma subjects and healthy children under the influence of general anaesthesia. The conclusion is on expected lines and would be of interest to the readers.

Author Response

Response to Reviewer 2 Comments

Point 1: The authors need to be commended for the manuscript.  It is well known that I-care tonometer tends to overestimate IOP as compared to GAT, especially at higher IOP values. The authors have compared intraocular pressure (IOP) measurements using rebound tonometry (iCare) and Perkins applanation tonometry in childhood glaucoma subjects and healthy children under the influence of general anaesthesia. The conclusion is on expected lines and would be of interest to the readers.

Response 1: Thank you very much for your praise of our work.

Reviewer 3 Report

well conducted, detailed work

line 286: further consideration is needed on the alleged overestimation of iCare compared to Perkins (Goldmann): recent studies have found that Goldmann tonometry underestimates true IOP . See and cite work by Messenio et al - Messenio, D.; Ferroni, M.; Boschetti, F. Goldmann Tonometry and Corneal Biomechanics. Appl. Sci. 2021, 11, 4025. https://doi.org/ 10.3390/app11094025)
and:
Takagi, D.; Sawada, A.; Yamamoto, T. Evaluation of a New Rebound Self-tonometer, Icare HOME: Comparison With Goldmann
Applanation Tonometer. J. Glaucoma 2017, 26, 613-618.

These considerations are important to avoid simplifying the discourse on the "overestimation" of iCare compared to Perkins.

Author Response

Response to Reviewer 3 Comments

Point 1: Line 286: further consideration is needed on the alleged overestimation of iCare compared to Perkins (Goldmann): recent studies have found that Goldmann tonometry underestimates true IOP . See and cite work by Messenio et al - Messenio, D.; Ferroni, M.; Boschetti, F. Goldmann Tonometry and Corneal Biomechanics. Appl. Sci. 2021, 11, 4025. https://doi.org/ 10.3390/app11094025)
and:
Takagi, D.; Sawada, A.; Yamamoto, T. Evaluation of a New Rebound Self-tonometer, Icare HOME: Comparison With Goldmann
Applanation Tonometer. J. Glaucoma 2017, 26, 613-618.

These considerations are important to avoid simplifying the discourse on the "overestimation" of iCare compared to Perkins.

Response 1: Thank you very much for helpful comments on our work. We added an extra paragraph to the discussion: ‘In contrary, there are studies which proven that both iCare and Goldmann tonometry underestimate the real IOP. According to Takagi at al. study the iCare tends to underestimate the IOP in comparison to Goldmann tonometry in lower IOP[38]. The reason for it was not given in this paper. The latest Messenio et al. study concerning the Goldmann tonometry described the underestimation of IOP by Goldmann tonometry due to thinner corneas, which was already mentioned in other study before[39,40]. We cannot support these statements with our data. Our IOP measurements were high from the beginning and the corneal thickness was normal to high.’